# Pharmacokinetics, Tissue Distribution, Plasma Protein Binding Studies of 10-Dehydroxyl-12-Demethoxy-Conophylline, a Novel Anti-Tumor Candidate, in Rats

**DOI:** 10.3390/molecules24020283

**Published:** 2019-01-14

**Authors:** Chengjun Jiang, Jie Li, Xianghai Cai, Nini Li, Yan Guo, Dianlei Wang

**Affiliations:** 1School of Pharmacy, Anhui University of Chinese Medicine, Hefei 230012, China; jcj7166@gmail.com (C.J.); 18256960482@sina.cn (J.L.); linini18756919080@126.com (N.L.); GY1689349984@163.com (Y.G.); 2School of Environmental and Chemical Engineering, Zhaoqing University, Zhaoqing 526061, China; 3State Key Laboratory of Phytochemistry and Plant Resources in West China, Kunming Institute of Botany, Chinese Academy of Sciences, Kunming 650201, China; xhcai@mail.kib.ac.cn

**Keywords:** 10-dehydroxyl-12-demethoxy-conophylline, pharmacokinetics, tissue distribution, protein binding rate

## Abstract

10-Dehydroxyl-12-demethoxy-conophylline is a natural anticancer candidate. The motivation of this study was to explore the pharmacokinetic profiles, tissue distribution, and plasma protein binding of 10-dehydroxyl-12-demethoxy-conophylline in Sprague Dawley rats. A rapid, sensitive, and specific ultra-performance liquid chromatography (UPLC) system with a fluorescence (FLR) detection method was developed for the determination of 10-dehydroxyl-12-demethoxy-conophylline in different rat biological samples. After intravenous (i.v.) dosing of 10-dehydroxyl-12-demethoxy-conophylline at different levels (4, 8, and 12 mg/kg), the half-life t_1/2α_ of intravenous administration was about 7 min and the t_1/2β_ was about 68 min. The AUC_0→∞_ increased in a dose-proportional manner from 68.478 μg/L·min for 4 mg/kg to 305.616 mg/L·min for 12 mg/kg. After intragastrical (*i.g*.) dosing of 20 mg/kg, plasma levels of 10-dehydroxyl-12-demethoxy-conophylline peaked at about 90 min. 10-dehydroxyl-12-demethoxy-conophyllinea absolute oral bioavailability was only 15.79%. The pharmacokinetics process of the drug was fit to a two-room model. Following a single i.v. dose (8 mg/kg), 10-dehydroxyl-12-demethoxy-conophylline was detected in all examined tissues with the highest in kidney, liver, and lung. Equilibrium dialysis was used to evaluate plasma protein binding of 10-dehydroxyl-12-demethoxy-conophylline at three concentrations (1.00, 2.50, and 5.00 µg/mL). Results indicated a very high protein binding degree (over 80%), reducing substantially the free fraction of the compound.

## 1. Introduction

10-Dehydroxyl-12-demethoxy-conophylline (Figure 1) isolated from *Melodinus tenuicaudatus* is an indole alkaloid consisting of two pentacyclic aspidosperma skeletons. Aspidosperma alkaloids, the largest alkaloid in the monoterpenoid indole alkaloid (MIA) family, can be extracted from a variety of plants, with approximately 250 family members [1]. Monoterpenoid indole alkaloids (MIAs) provide a rich source of medicinal products. Examples include the anticancer compounds vinblastine and vincristine from *Catharanthus roseus* (Madagascar periwinkle) and camptothecin from *Camptotheca acuminata* [2]. Monoterpenoid indole alkaloids were brought to the attention of many researchers due to the MIAs’ complicated structures and potent biological activities [3,4]. Aspidosperma alkaloid bases are anti-parasitic, including Plasmodium, Leishmania, and Trypanosoma, and have potential cytotoxicity that can be used to develop anti-tumor effects [5]. Researchers Cai et al. [6,7] found strong anti-tumor cytotoxic components in the study of indole alkaloid anti-tumor activity molecules. Based on the structure–activity relationship analysis, it was found that aspidospermine and 10-dehydroxyl-12-demethoxy-conophylline represent one of the basic skeletons of anti-tumor activity in active monoterpene alkaloids [8]. At present, the research on 10-dehydroxyl-12-demethoxyconophylline focuses on chemical composition and pharmacological activity [6,7,9]. It has significant inhibitory activity on five human cancer cell lines, namely, the HL-60, SMMC-7721, A-549, MCF-7, and SW480 tumor cell lines [10]. Further mechanism studies showed that the active ingredient of 10-dehydroxyl-12-demethoxy-conophylline exerts its anti-tumor activity through the inhibition of the Wnt signaling pathway (IC_50_ = 0.21, 0.45 μm) [11]. This is completely different from the structure and mechanism of the tubulin inhibitor vinblastine components [12,13] suggesting better selectivity and less toxic effects on tumor cells. Among the candidate compounds of the new drug, and based on pharmacodynamic and toxicity characteristics, it is of great significance to thoroughly study the pharmacokinetic properties of 10-dehydroxyl-12-demethoxy-conophylline.

According to previous pharmacodynamic and toxicity results [14], the candidate compound has the above two characteristics, and is suitable as a candidate compound for further pharmacokinetic study, in order to conduct pharmacokinetic screening and evaluation. To our knowledge, no ultra-performance liquid chromatography–fluorescence (UPLC-FLR) method for the quantification of pharmacokinetic profiles and distribution characteristics of 10-dehydroxyl-12-demethoxy-conophylline in rats has been performed. Therefore, in this study, the plasma pharmacokinetic parameters, tissue distribution, and protein binding rate in rat and human plasmas and in rats were studied using the UPLC-FLR method.

## 2. Results and Discussion

### 2.1. Validation of Analytical Method

The method was validated in terms of selectivity, linearity, precision, accuracy, recovery, and stability according to the European Medicines Agency’s guidelines [15]. The calibration curve of 10-dehydroxyl-12-demethoxy-conophylline in rat blood samples was linear, ranging from 39 to 6000 ng/mL (39, 78, 156, 310, 1250, 2500, 5000, 6000 ng/mL) with a lower limit of quantification (LLOQ) of 39 ng/mL. R^2^ for all standard curves was 0.9973. The relative standard deviation of precision was 4.51%, 8.29%, 2.55%, and 1.46% at 39, 156, 1250, and 4000 ng/mL, respectively. Accuracies determined for intra-day and inter-day were all within 100% ± 15% of the actual values. Moreover, no significant matrix effect was observed after dilution. Thus, this optimized UPLC method was proven to be sensitive, selective, and rapid for the determination of 10-dehydroxyl-12-demethoxyconophylline in rat plasma.

The calibration curve of 10-dehydroxyl-12-demethoxy-conophylline were prepared by spiking the standard working stock solutions with 10μL of the different concentration, and 120μL blank rat tissue homogenate sample in a 1.5 mL centrifuge tube. The calibration standards were prepared at concentrations of 0.292, 0.586, 1.172, 2.343, 4.688, 9.375, and 18.751 μg/mL for the tissue sample. With the concentration (C) as the abscissa, the peak area of the analyte was used as the ordinate for linear regression. The lower limit of quantitation was determined according to the signal-to-noise ratio (S/N) of the chromatogram, and the lower limit of detection was determined according to the (S/N) of 3. We used the weighted least squares method to perform the regression operation to obtain the linear regression equation, which is the standard curve. The calibration curves, correlation coefficients, and linear ranges of 10-dehydroxyl-12-demethoxy-conophylline in plasma and tissue are shown in Table 1.

### 2.2. Pharmacokinetics Study

The plasma concentration–time data were fitted using the DAS 2.0 program. The concentration–time curves of 10-dehydroxyl-12-demethoxy-conophylline following intragastrical and intravenous administration are shown in Figure 2 and Figure 3 and Table 2.

After i.v. administration of 10-dehydroxyl-12-demethoxy-conophyllinein in rats, the elimination half-life was approximately 7 min and the t_1/2β_ was about 68 min. AUC_0→∞_ increased in a dose-proportional manner from 68.478 μg/L·min for 4 mg/kg to 305.616 mg/L·min for 12 mg/kg. After i.g. dosing at 20 mg/kg, plasma levels of 10-dehydroxyl-12-demethoxy-conophylline peaked at about 90 min. 10-Dehydroxyl-12-demethoxy-conophylline absolute oral bioavailability was only 15.79%. The experimental results show a good linear correlation between AUC_0→t_ and the dose studied (*r* = 0.9821), as shown in Figure 4, suggesting that the in vivo process of 10-dehydroxyl-12-demethoxy-conophylline conforms to linear kinetics over the dose range studied. Characteristics, analysis of variance for t_1/2_, CL, and V were not significantly different (*p* > 0.05).

At present, there are no studies on the pharmacokinetics of 10-dehydroxyl-12-demethoxyconophylline in China. The pharmacokinetic profiles of a single injection experiment in the tail vein of rats can be found initially, and the drug is consistent with a two-compartment pharmacokinetic model. According to the characteristics of the two-compartment model drug, the drug has the fastest distribution in the central chamber with high blood perfusion and a slower entry into the outer tissue compartment. The pharmacokinetic profiles show that t_1/2α_ is only about 7 min and t_1/2β_ is about 69 min, which means that the elimination is very fast in the central compartment and the elimination of the tissue is relatively slow.

### 2.3. Tissue Distribution Study

Tissue distribution was assessed at three different time points (5, 20, and 45 min) after intravenous administration of 8 mg/kg 10-dehydroxyl-12-demethoxy-conophylline in rats. The results are presented schematically in Figure 5.

The results demonstrated that 10-dehydroxyl-12-demethoxy-conophylline underwent a rapid and wide distribution to all the examined tissues. Levels of 10-dehydroxyl-12-demethoxyconophyllinein in the heart and kidney were markedly higher than in other tissues at the 5 min time points. The concentration of the drug in tissues with high blood flow may be affected by the rate of blood perfusion. At the equilibrium time of 20 min, the concentration of drug mainly distributed in the liver, kidney, and lung tissues gradually decreased, and the drug concentration in the lung reached the highest at this time, which may be related to the accumulation of drugs in the lungs. At the elimination period of 45 min, it mainly distributed in the liver and kidney tissues, and the drugs in each tissue gradually decreased to a very low concentration. It can be seen from the data that the drug is mainly distributed in tissues with large blood flow, and the distribution in the kidney and lung is the highest, indicating that the kidney and the lung are the target organs for the drug concentration of the drug [16]. During the course of drug administration from tail vein to elimination, the mean tissue-to-plasma ratios (Kp) of 10-dehydroxyl-12-demethoxy-conophylline increased at first and then decreased, which was consistent with the trend of tissue concentration, indicating that the distribution of drugs in tissues was mainly affected by blood flow (Figure 6). The concentration of plasma in the brain was kept to a small amount, which may be affected by blood–brain barrier factors, which provide a great research direction and value for the practical application of the drug. Therefore, the tissue distribution experiment provides a reference for the study of the in vivo distribution characteristics of drugs and provides a theoretical basis for the development of drugs in the later stage.

### 2.4. Protein Binding Study

Equilibrium dialysis is considered a reference method [17] for assessing plasma protein–drug binding rate where non-specific binding has no impact on the determination of the free fraction [18]. Protein binding degrees evaluated using three concentration levels (1.00, 2.50, and 5.00 µg/mL) of 10-dehydroxyl-12-demethoxy-conophylline were 81.86% ± 4.2%, 82.61% ± 3.1%, and 81.25% ± 2.9%, in rat plasma (Figure 7), and were 86.88% ± 7.3%, 84.10% ± 2.7%, and 84.65% ± 6.1%, in human plasma (Figure 8), respectively.

This experimental study shows that 10-dehydroxyl-12-demethoxy-conophylline is more significantly combined with the protein to form a binding drug in the plasma, and there is no significant difference in the protein binding rate between the rat and the human, after the drug enters the circulatory system. When 10-dehydroxyl-12-demethoxy-conophylline is present at the same time as a drug with a competitive binding protein, it may cause an increase in the concentration of free drug in the plasma to cause a poisoning reaction. It is necessary to consider this factor in subsequent drug interaction study experiments, in order to ensure an effective drug concentration.

## 3. Materials and Methods

### 3.1. Chemicals and Reagents

10-Dehydroxyl-12-demethoxy-conophylline (purity >99.0%) was provided by the Kunming Institute of Botany, Chinese Academy of Sciences. Internal standard 11-methoxymelodinine (purity >99.0%) was provided by the Kunming Institute of Botany, Chinese Academy of Sciences. Methanol (chromatographically pure) was purchased from OCEANPAK Chemicals (Baotian Technology Biological Co., Ltd., Hefei, China) Normal saline was purchased from Beijing Jiajia Kangtai Trading Co., Ltd. (Beijing, China). All other reagents and chemicals were analytical reagents and did not require further purification.

### 3.2. Animal Experimentation

Sprague Dawley rats (half male and female) with a body mass 180–220 g were provided by the Laboratory Animal Center of Medical University of Anhui (Hefei, China), Certificate number: SCXK(wan) 2015-002. Animals were housed under controlled conditions (temperature 22 ± 2 °C, humidity 50% ± 5%, 12 h dark/light cycle) with standardized diet and acclimatized for 7 days, prior to the experiments. All protocols and care of the rats were performed in strict compliance with the Guidelines for the Use of Laboratory Animals (National Research Council) and authorized by the Animal Care and Use Committee of Anhui University of Chinese Medicine.

### 3.3. UPLC-FLR Analysis

Chromatographic separation was performed using a UPLC system (Waters ACQUITY-CLASS. US) with an analytical column Kinetex (C18, 1.7 μm, 2.1 mm × 100 mm; Phenomenex; California; CA, USA) at 25 °C. Water and methanol (75:25) were used as the isocratic elution, running at a flow rate of 0.2 mL/min. The detector was used with fluorescence (FLR) detection with a fluorescence excitation wavelength of 330 nm and a fluorescence emission wavelength of 439 nm.

### 3.4. Pharmacokinetic Studies

The rat intravenous doses were determined as 4, 8, and 12 mg/kg, and the intragastric administration dose was 20 mg/kg. These were selected to investigate whether the pharmacokinetics of the drug was linear over the range of doses used. In all, 18 (SD) rats (half male and female) were randomly divided into three groups, namely, high-, medium- and low-dose groups. The rats fasted 12 h be before the administration of 10-dehydroxyl-12-demethoxy-conophylline but with free access to water. 10-dehydroxyl-12-demethoxy-conophylline was dissolved in normal saline and added with 5% ethanol. Amounts of 4, 8, and 12 mg/kg (0.5 mL/100 g) were injected into the tail vein of the rat, and 0.3 mL of blood was taken from the eyelid before administration (0 min) and 2, 5, 15, 30, 60, 90, 120, 180, 240, and 300 min after intravenous administration. Another 6 rats (half male and female) were administered an intragastrical (*i.g.*) dose of 20 mg/kg. Blood samples (0.3 mL) were collected by retro-orbital bleeding into heparinized 1.5mL tubes before administration (0 min) and 15, 30, 60, 90, 120, 240, 480, and 600 min after intragastric administration. Immediately after collection, the blood samples were centrifuged at 3500 rpm for 10 min to obtain plasma, and originating supernatants were also stored at −70 °C until analysis.

Plasma samples (90 μL) were spiked with 10 μL internal standard (100 ng/mL) and extracted with 300 μL of methanol by vortexing for 10 min. The plasma and aqueous phases were separated by centrifugation at 14,000 rpm for 10 min. The supernatants were transferred to another tube and evaporated to dryness at 50 °C. The residue was dissolved in 100 μL methanol and vortexed. A 2 μL aliquot of the solution was injected into the UPLC system for analysis.

### 3.5. Tissue Distribution Study

For the tissue concentration study of 10-dehydroxyl-12-demethoxy-conophylline, 18 (SD) rats (half male and female) were randomly divided into three groups. The rats were administered intravenously at a dose of 8 mg/kg of 10-dehydroxyl-12-demethoxy-conophylline, and the rats were euthanized at 5, 20, and 45 min after administration. Samples of their tissues and blood were carefully collected and washed with sodium phosphate buffer (pH 7.4) to remove blood. Each tissue (0.2 g) was homogenized in normal saline (1 mL). Subsequently, an amount 120 μL of the homogenate was pipetted into 480 μL of methanol and vortexed for 3 min, and then centrifuged at 14,000 rpm for 10 min. The supernatant was then centrifuged again at 14,000 rpm for 10 min and stored at −70 °C until analysis. The preparation process for analysis was the same as described above for plasma.

### 3.6. Plasma Protein Binding Test

Protein binding was determined in vitro by equilibrium dialysis of plasma of rats or human serum albumin. Blank dialysate, internal dialysate, and external dialysate were prepared separately. The internal dialysate was prepared by taking 0.4 mL of blank plasma and adding 100 μL of the 10-dehydroxyl-12-demethoxy-conophylline series standard solution to prepare the equivalent of 10-dehydroxyl-12-demethoxy-conophylline plasma concentrations of 0.078, 0.156, 0.312, 0.625, 1.250, 2.500, 5.000, and 20.000 μg/mL simulating biological samples. The external dialysate was prepared by taking 0.4 mL of blank dialysate and adding 100 μL of the 10-dehydroxyl-12-demethoxy-conophylline series standard solution to prepare the equivalent of 10-dehydroxyl-12-demethoxyconophylline plasma concentrations of 0.078, 0.156, 0.312, 0.625, 1.250, 2.500, 5.000, and 20.000 μL/mL simulating dialysate samples. The protein binding rate of 10-dehydroxyl-12-demethoxy-conophylline in plasma was measured using the equilibrium dialysis method in this experiment. The concentration of unbound compound (B) and bound compound to protein (A) were calculated as follows:F = (A − B)/A,(1)
where F is the bounded fraction, A is the compound concentration in the plasma compartment, and B is the concentration of the compound in the phosphate buffer compartment.

### 3.7. Pharmacokinetic and Statistical Analysis

The pharmacokinetic parameters were calculated using the pharmacokinetic software DAS 2.0 (Chinese Pharmacological Association, Beijing, China) by non-compartmental and compartmental analysis method. The elimination half-life (t_1/2_) was determined by linear regression of the terminal portion (last the points) of the plasma concentration–time curve. The area under the plasma concentration–time curve from zero to the last measurable plasma concentration point (AUC_0-t_) was calculated using the linear trapezoidal method. Extrapolation from time zero to infinity (AUC_0-∞_) was calculated as follows:AUC_0-∞_ = AUC_0-t_ + C_t_/ke,(2)
where C_t_ is the last measurable plasma concentration and ke is the terminal elimination rate constant. Absolute oral bioavailability (F) of 10-dehydroxyl-12-demethoxy-conophylline was calculated according to the following equation:(3)$$F\;\left(\%\right)=\frac{{\mathrm{AUC}}_{\left(\mathrm{ig}\right)}/{\mathrm{Dose}}_{\left(\mathrm{ig}\right)}}{{\mathrm{AUC}}_{\left(\mathrm{iv}\right)}/{\mathrm{Dose}}_{\left(\mathrm{iv}\right)}}\times 100,$$
where AUC_ig_ and AUC_iv_ are the AUC values after intragastric and intravenous administration of 10-dehydroxyl-12-demethoxy-conophylline, respectively, and D_iv_ and D_ig_ are the doses for intravenous and intragastric administration of 10-dehydroxyl-12-demethoxyconophylline, respectively. Data were presented as means with their standard deviation (mean ± SD).

## 4. Conclusions

Solid pharmacokinetic knowledge of natural compounds can play a critical role in proper design and choice of in vitro pharmacological models to evaluate the molecular mechanisms in a kinetically relevant manner [19,20,21]. In this paper, a rapid, sensitive, and specific UPLC-FLR method was developed for the determination of 10-dehydroxyl-12-demethoxy-conophylline in different rat biological sample. To the best of our knowledge, this paper is the first report of a thorough study of the pharmacokinetic profiles, distribution characteristics, and protein binding rate of 10-dehydroxyl-12-demethoxyconophylline in rats. This work lays a theoretical foundation for further pharmacological research and has certain new drug development value.

## Figures and Tables

**Figure 1 molecules-24-00283-f001:**
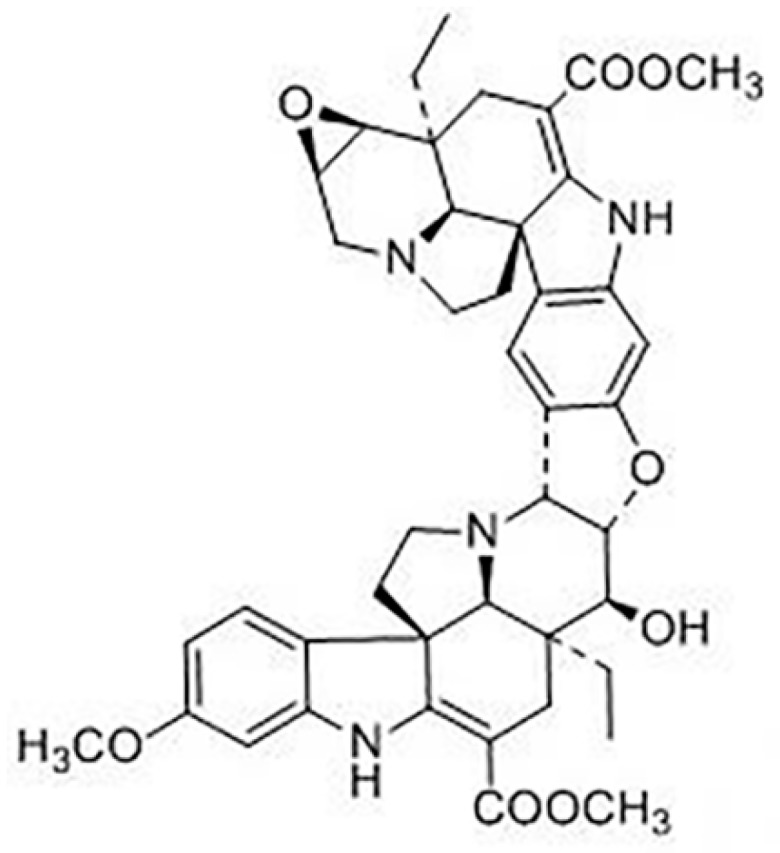
Chemical structure of 10-dehydroxyl-12-demethoxy-conophylline.

**Figure 2 molecules-24-00283-f002:**
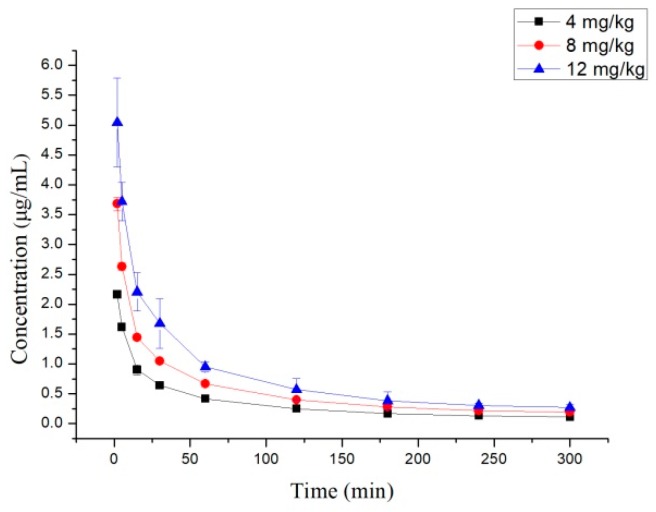
Mean plasma concentration–time (mean ± SD) profiles of 10-dehydroxyl-12-demethoxyconophylline in rats after receiving a single intravenous (i.v.) dose at different concentration levels (4, 8, and 12 mg/kg, *n* = 6).

**Figure 3 molecules-24-00283-f003:**
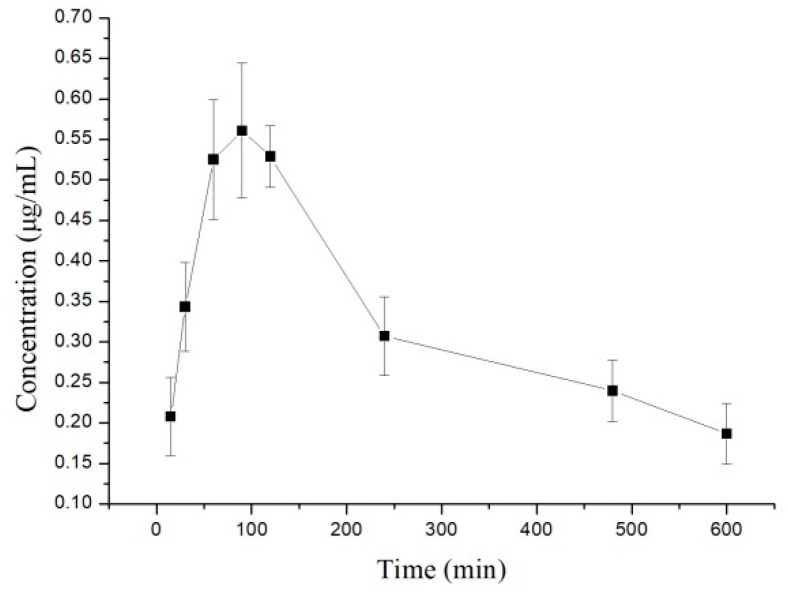
Mean plasma concentration–time (mean ± SD) profiles of 10-dehydroxyl-12-demethoxyconophylline following an intragastrical (i.g.) dose of 20 mg/kg in rats (*n* = 6).

**Figure 4 molecules-24-00283-f004:**
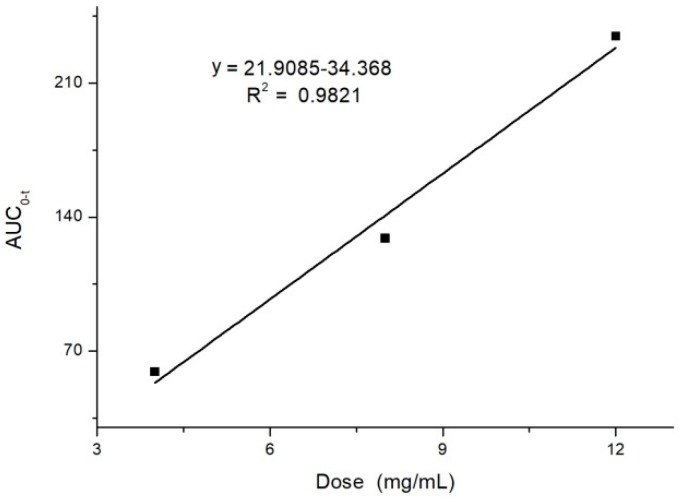
Dose positive correlation of single dose (4, 8, and 12 mg/kg) AUC_0→t_ (*n* = 6).

**Figure 5 molecules-24-00283-f005:**
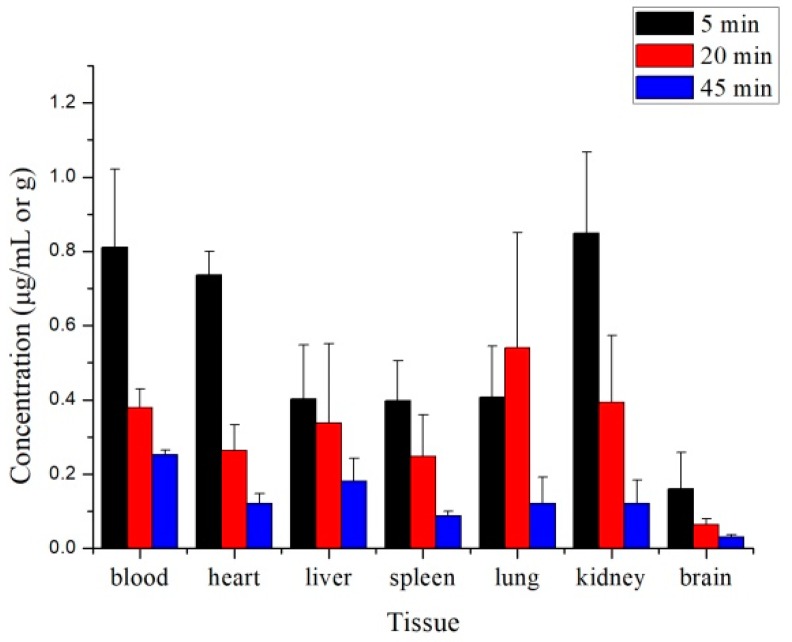
Concentration of 10-dehydroxyl-12-demethoxy-conophylline in rat tissues at 5, 20, and 45 min after receiving a single i.v. dose of 10-dehydroxyl-12-demethoxy-conophylline at 8 mg/kg (mean ± SD, *n* = 6).

**Figure 6 molecules-24-00283-f006:**
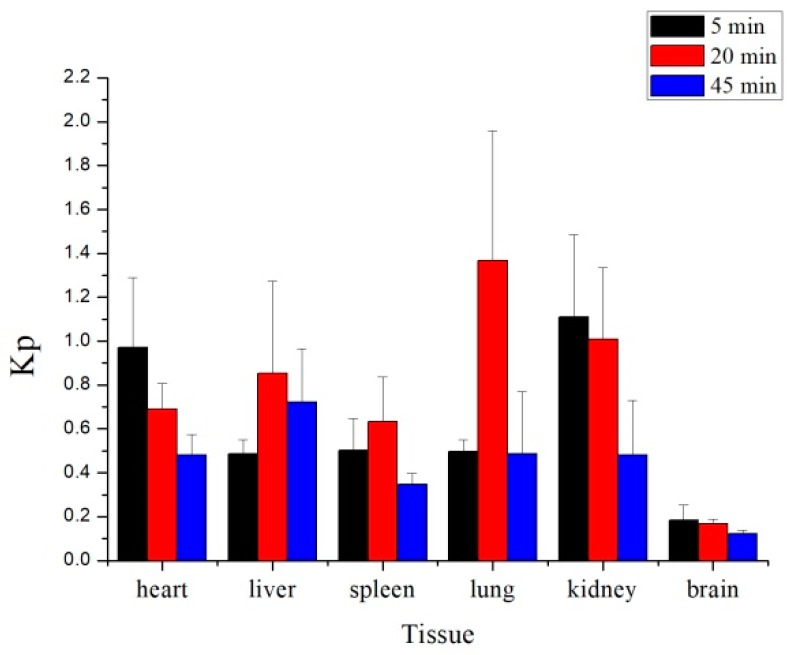
Ratio of tissue-to-plasma concentrations (Kp) at 5, 20, and 45 min after receiving a single i.v. dose of 10-dehydroxyl-12-demethoxy-conophylline at 8 mg/kg (mean ± SD, *n* = 6).

**Figure 7 molecules-24-00283-f007:**
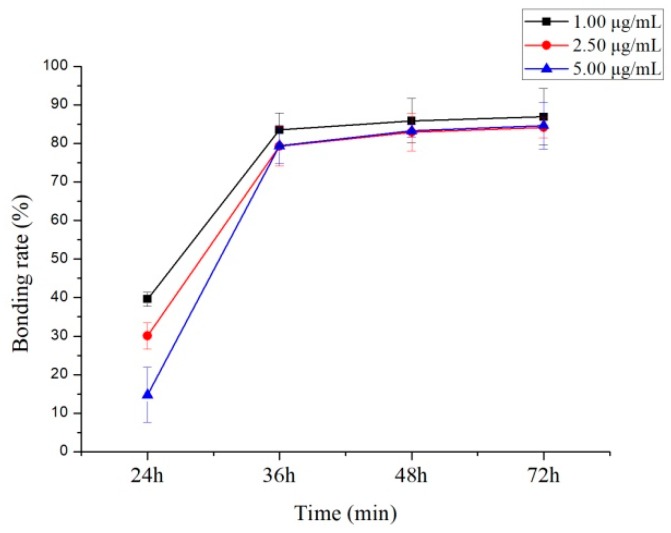
Protein binding ratio of 10-dehydroxyl-12-demethoxy-conophylline in rat plasma.

**Figure 8 molecules-24-00283-f008:**
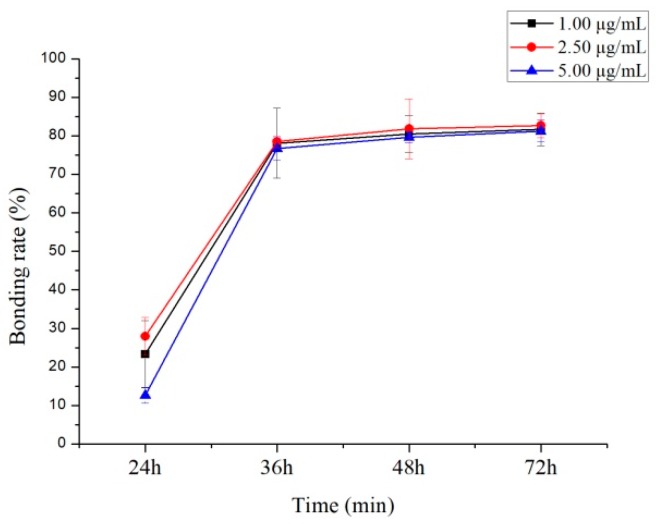
Protein binding ratio of 10-dehydroxyl-12-demethoxy-conophylline in human plasma.

**Table 1 molecules-24-00283-t001:** The calibration curves, correlation coefficients, and linear ranges of 10-dehydroxyl-12-demethoxy-conophylline in plasma and tissue are shown in the table.

Sample	Calibration Curve	R^2^	Linear Range (μg/mL)
Plasma	y = 1.3661 + 0.008	R^2^ = 0.9973	0.039–6.000
Heart	y = 8577.8x − 1701.100	R^2^ = 0.9977	0.29–18.700
Liver	y = 7997.6x − 3192.600	R^2^ = 0.9956	0.292–9.375
Spleen	y = 8203.4x − 164.790	R^2^ = 0.9992	0.292–18.780
Lung	y = 7602.5x + 911.940	R^2^ = 0.9910	0.292–9.375
Kidney	y = 8413.3x − 194.040	R^2^ = 0.9970	0.292–18.700
Brain	y = 9592.8x − 2367.300	R^2^ = 0.9943	0.292–9.375

**Table 2 molecules-24-00283-t002:** Profiles of 10-dehydroxyl-12-demethoxy-conophylline after oral administration of 20 mg/kg and intravenous administration of 4, 8, and 12 mg/kg in rats (mean ± SD, *n* = 6).

Parameter	4 mg/kg	8 mg/kg	12 mg/kg	20 mg/kg
	(i.v.)	(i.v.)	(i.v.)	(i.g.)
t_1/2α_ (min)	6.983 ± 2.299	7.108 ± 1.908	7.046 ± 2.050	52.415 ± 1.920
t_1/2β_ (min)	68.529 ± 3.290	66.465 ± 4.551	69.315 ± 4.324	69.942 ± 2.680
T_max_ (min)	2 ± 0	2 ± 0	2 ± 0	90 ± 0
CL (mL/min)	0.041 ± 0.070	0.039 ± 0.120	0.041 ± 0.050	0.046 ± 0.013
V (L/kg)	2.096 ± 0.0465	1.906 ± 0.012	2.014 ± 0.019	4.881 ± 2.390
MRT (min)	91.444 ± 4.428	85.105 ± 1.882	81.871 ± 5.157	256.76 ± 3.510
AUC_0–t_ (μg·min/mL)	59.256 ± 6.910	128.938 ± 8.871	234.524 ± 12.070	52.048 ± 4.671
AUC_0–∞_ (μg·min/mL)	68.478 ± 8.167	131.438 ± 9.392	305.616 ± 17.432	74.307 ± 16.548

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
