# Peer review of "Pharmacokinetics, Tissue Distribution, Plasma Protein Binding Studies of 10-Dehydroxyl-12-Demethoxy-Conophylline, a Novel Anti-Tumor Candidate, in Rats"

_molecules, 2019, doi:10.3390/molecules24020283_

Round 1

Reviewer 1 Report

In the paper, authors described Pharmacokinetics, Tissue Distribution, Plasma Protein Binding Studies of 10-dehydroxyl-12-demethoxy-conophylline,a novel anti-tumor candidate, in Rat.

 Majors corrections and some minors are necessary beforefinal acceptation

Introduction :

- For me references 5 and 6 do not include the researcher Cai Xiang Hai, please check

-What is the meaning of the abbreviation Wnt line 49

-What is the meaning of the abbreviation FLR line 57

- Line 54 authors must include references  after the sentence "According to the previous pharmaconynamics and toxicity results.." ref?

Material and methods

-section 2.2: The authors must indicate the agreement number they have obtained for this animal experimentation on the part of their authority

-section 2.3:Do the authors use an internal standard in their chromatographic method? if yes, which one? if no, why they don't use one?

-section 2.4: In this paragraph the authors indicate that the dose of 20 mg/. KG is administered by IV. It is not described as a method of administration of this dose by gavage while in the bet results, the authors indicate results obtained by gavage in table 2 and for the calculation of F
Authors must clearly indicate the administration protocol by gavage

- ection 2.4: The authors do not cleary describe how are prepared the plasma samples before injection in the chromatographic system.The authors need to clarify this point.

- section 2.7: Line 135, what does the term Wogonin

-section 2.7; Line 137. The authors have to check  formula of F. For me, there is an error at the AUC. AUCig is used instead to AUCiv and conversely

Result  and discussion

In the title of this paragraph, i think that the authors mut use "Results" and no "Result"

-section 3.1: The authors indicate in the text that plasma linearity ranges from 39 to 6000 ng/ml. Why in table I the range of plasma linearity is indicated only from 39 to 2500 ng/ml?
Authors should also use the same units between text and table

-section 3.1: The authors are required to provide chromatograms for the plots obtained for a blank, a calibration point, a plasma sample and a tissue sample

-section 3.1: Line 149, the authors give the RSD values for the concentration of 39, 156, 1250, and 4000 ng/ml. The first 3 concentrations correspond to calibration point concentrations. What is the concentration 4000 ng/ml
The authors have used quality controls, if so they must give the concentrations which must be different from those of the calibration points.

-section 3.2 : The figure 2 and 3 are too small, difficult to read and must be enlarged

- section 3.2 : Figure 3 shows the profile of concentrations over time after administration of 100 mg/kg  dose by gavage.
The use of this dose has never been described in animal treatment protocols. The authors need to clarify this point.

-section 3.2: Lines 180-188. I do not understand why there is a proportionality between doses and AUC0-t and not between doses and AUC0-inf. The authors need to clarify this point.

-section 3.3:The caption in Figure 5 does not match what is shown on the figure

-section 3.3: Figure 5, the term  "blood " must be replaced by  "Plasma

- section 3.3: Line 120. Why did the authors choose as tissue sampling time, 5, 20 and 45 min? What are the criteria of their choice?
Why did the authors not choose a very late sampling time of 300 min corresponding to the last point of plasma kinetics?

Author Response

Thank you for your kind suggestions. Your suggestions and comments are valuable for us. We have carefully considered the comments and have accordingly revised the manuscript. Attached please see response.

Reviewer 2 Report

In this study, authors reported Pharmacokinetics, Tissue Distribution, Plasma Protein Binding Studies of 10-dehydroxly-12-demethoxy-conophylline, a novel anti-tumor candidate, in Rat.Overall, this study givesinteresting information about natural anticancer candidate. However, some issues have to be clarified or fixed before publication.

        Major Comments:

Please check the title, sub-title, font size, format, spelling, space write, capital, small letter, and reference according to the Author guideline. Please check all of them as a whole.

In this study, there is no match between methods and results. Please accurate method and result the each other. Please correct again by referring to below.

In 2.4. The pharmacokinetics of 10-dehydroxyl-12-demethoxy-conophylline in rats

 - Author mentioned that 4, 8, and 12 mg/kg were injected into the tail vein. However, the legend of Figure 2, which showed plasma concentration-time data after IV, was referred to as 10, 20, and 40 mg/kg.

- Author mentioned that another 6 rats were administered intravenously with 20 mg/kg. However, 3.2. The pharmacokinetics of 10-dehydroxy-12-demeth-oxy-conophylline in rats section, it was referred that author i.g. (intragastrical) administrated 20 mg/kg to rats, and dose was not match with legend of figure 3, and in the legend of Table 2., It was referred to oral administration.

In 2.5. Experimental Study on Tissue Distribution of 10-dehydroxyl-12-demethoxyconophtlline in Rats

- For the Experimental Study on Tissue Distribution, please present more information to Method the tissue collected.

In 2.7 Pharmacokinetic and Statistical Analysis

- Author referred to pharmacokinetics method by non-compartmental, but the Table 2 has t1/2 divided into t1/2α and t1/2β. There is a two-compartment model methods parameter. It is recommended to revise ‘non-compartmental’ to ‘non-compartmental and compartmental analyses’ in the method section.

In 3.3. Experimental Study on Tissue Distribution of 10-dehydroxyl-12-demethoxyconophtlline in Rats

- Author referred that livers and kidney were mainly distributed tissue. I think that the concentration and Kp of Lung are also main distribution tissue. Therefore, the author should write in the discussion section why author suggested only liver and kidney as a mainly organ.

Minor Comments:

Generally, it is recommended to abbreviate the 10-dehydroxly-12-demethoxy-conophylline, which is stated as a natural anticancer candidate. As there are some parts of the text that are not properly written about the 10-dehydroxly-12-demethoxy-conophylline. Therefore, please check spelling of 10-dehydroxyl-12-demethoxy-conophylline. (e.g. In line 16, 17, 20, 21, 99, 100, 117, 174, 191, and 228)

Check the effective number of data and unify the decimal places.

Author Response

Dear reviewer       

      On behalf of my co-authors, we thank you very much for giving us an opportunity  to revise our manuscript. We appreciate reviewers very much for your positive and constructive comments and suggestions on our manuscript.

      We have studied reviewer’s comments carefully and have made revision which marked in red in the paper. We have tried our best to revise our manuscript according to the comments. Attached please see the response.

      We would like you to express our great appreciation to you for comments on our paper. 

Round 2

Reviewer 1 Report

After proofreading the corrected version, the authors proceeded to corrections and introduced additional information
Under these conditions, I agree to publish this corrected version

Author Response

      Thanks very much for your kind word and consideration on revise paper of my paper. On behalf of my co-authors, we would like to express our great appreciation to reviewers and editors.

Reviewer 2 Report

The manuscript "Pharmacokinetics, Tissue Distribution, Plasma Protein Binding Studies of 10-dehydroxyl-12-demethoxy-conophylline, a novel anti-tumor candidate, in Rats" was revised as reviewer’s comments. Some parts are reflected according to the comments, but author should have checked revised manuscript according to the author guideline again. Author need to compensate this.

- The composition order of revised manuscript was not match with author guideline.        Please recheck author guideline.

- Please check the position of figure legends. (e.g. Figure 2 and 3)

- Please check the sub-title. (e.g. In line 99, 108, 126, 166, 199 and 228)

Author Response

1. Comment: The composition order of revised manuscript was not match with author guideline. Please recheck author guideline.

Response:  Thank you very much. According to your suggestion, the composition order of revised manuscript was adjusted by reference to the author's guide. (eg.  Introduction, Results, Discussion, Materials and Methods, Conclusions).

2. Comment: Please check the position of figure legends. (e.g. Figure 2 and 3)

Line 95, 100 in revise paper

Response: We sincerely apologize for our thoughtless. According to your kindly suggestion, we have carefully checked the position of figure legends,correction has been made in the revised version.

3. Comment: Please check the sub-title. (e.g. In line 99, 108, 126, 166, 199 and 228) .

Response: Thank you very much. According to your suggestion, we have carefully checked the sub-title. We changed the corresponding sub-title (e.g. in line 90, 125,156,197, 219, 230,)